# Comparative Analysis of the Efficacy of Different Regimens of 12 Months Rifaximin-Alfa Therapy in Patients with Liver Cirrhosis and Minimal Hepatic Encephalopathy

**DOI:** 10.3390/diagnostics13203239

**Published:** 2023-10-17

**Authors:** Igor G. Bakulin, Kristina N. Ivanova, Elena Y. Eremina, Natalya V. Marchenko

**Affiliations:** 1Department of Propaedeutics of Internal Medicine, Gastroenterology and Dietetics Named after S.M. Ryss «Mechnikov North-Western State Medical University» of the Ministry of Health of Russia, 191015 St. Petersburg, Russia; 2Department of Propaedeutics of Internal Diseases, Federal State Budgetary Educational Institution of Higher Education «National Research Mordovian State University Named after. N.P. Ogarev», 430005 Saransk, Russia; 3Clinical and Educational Center, Gastroenterology and Hepatology, St. Petersburg State University, 199034 St. Petersburg, Russia

**Keywords:** liver cirrhosis, hepatic encephalopathy, minimal hepatic encephalopathy, rifaximin-α

## Abstract

It is a matter of current interest which rifaximin-α regimens in patients with liver cirrhosis and minimal hepatic encephalopathy are the most efficient. Study objective: to evaluate the effect of various rifaximin-α regimens for 12 months on clinical and laboratory parameters and quality of life in patients with liver cirrhosis and minimal hepatic encephalopathy. Methods. It was a multicenter, prospective, open-label, observational study that included 288 patients with liver cirrhosis and minimal hepatic encephalopathy of both sexes over the age of 18 years, who were prescribed a 12-month course of treatment with rifaximin-α in accordance with the product label. Statistical analysis was performed in the population of patients who completed all visits according to the protocol (*n* = 258). Retrospectively, the patients were divided into two subgroups: subgroup 1 (continuous course)—patients who received the study drug for a year and the number of days of administration was 360 days (*n* = 41); subgroup 2 (cyclic course)—patients who received the study drug during the year for less than 360 days (*n* = 217). At each of the 4 visits, the quality of life was assessed using the CLDQ questionnaire, the time to perform the number connection test, the severity of symptoms associated with hepatic encephalopathy, and laboratory parameters. Results. During the 12-month observation period, an increase in the total score on the CLDQ quality of life questionnaire in patients with chronic liver diseases was revealed, which indicates an improvement in the quality of life of patients receiving rifaximin-α therapy. When patients were divided into subgroups depending on the duration of therapy, some benefits of continuous rifaximin-α therapy were noted in the more pronounced dynamics of decrease in the time to perform the number connection test, and in decreased severity of the following symptoms associated with hepatic encephalopathy: impaired concentration and memory, cognitive impairment, and decreased performance. Laboratory findings showed positive dynamics in both subgroups. Conclusion. A continuous rifaximin-α regimen in patients with liver cirrhosis and minimal hepatic encephalopathy for 12 months was superior to cyclic use with a more pronounced effect on the quality of life of patients and on the symptoms associated with hepatic encephalopathy.

## 1. Introduction

Liver cirrhosis (LC) is one of the most common conditions in the structure of digestive system morbidities in terms of mortality, as well as in terms of development of serious complications, such as hepatic encephalopathy, oesophageal and gastric varices hemorrhage, spontaneous bacterial peritonitis and hepatorenal syndrome, which urges the search for efficient methods to prevent disease progression and the development of complications [1].

Hepatic encephalopathy (HE) is a common and serious complication of HC resulting from hepatic failure and/or portosystemic shunt. HE is a potentially reversible syndrome characterized by a range of neuropsychiatric disorders resulting from the accumulation of neurotoxic substances in the bloodstream and, ultimately, in the brain. The occurrence of HE indicates an unfavorable course of HC and leads to more frequent hospitalization of patients and an increased risk of mortality [2]. The mechanisms that cause brain dysfunction in liver failure are still not fully understood. Most often, HE is associated directly with decreased ammonia metabolism. Ammonia is produced by enterocytes from glutamine due to bacterial catabolism of nitrogenous compounds in the colon. The intact liver neutralizes the ammonia of the portal vein, converting it into glutamine and preventing it from entering the systemic circulation. However, with advanced liver disease, complete neutralization of ammonia does not occur and its level in the blood rises [3].

Recent work has highlighted the synergistic effect of hyperammonemia and systemic inflammation. One study found that only patients with systemic inflammation signs of systemic inflammatory response syndrome, and/or elevated levels of proinflammatory cytokines (tumor necrosis factor-a [TNF-a], interleukin-6 [IL-6]) developed HE in the presence of hyperammonemia. Hyperammonemia results not only from increased ammonia production by enterocytes in the intestine, but also from liver failure responsible for decreased urea cycle function and/or the presence of portosystemic shunting. While the presence of systemic inflammation is clearly established in the pathogenesis of HE, the existence and nature of neuroinflammation is less well understood. According to this theory, activation of microglia is expected, which may be associated with other pathophysiological mechanisms. Increased glutamine levels associated with neuroinflammation result in increased glutamatergic and GABAergic tone, leading to neurological deterioration [4].

According to the severity of HE’s manifestations, it is divided into minimal and overt (class I–IV) according to West Haven’s criteria. Minimal hepatic encephalopathy (MHE) is the mildest form of this condition. MHE is defined as HE without symptoms on clinical or neurological examination, but with abnormal performance of psychometric tests, impaired working memory efficiency and psychomotor and visual–spatial abilities. MHE is associated with impaired driving skills and an increased risk of traffic accidents, as well as an increase in hospitalizations and mortality. According to experts, a gold-standard test for the diagnosis of MHE does not currently exist; however, it is believed that a combination of two neuropsychological tests and/or a neurophysiological test may be the standard for diagnosing MHE [5]. Overt HE shows abnormal blood ammonia levels and neurological symptoms, including asterixis, worsening of neurological and mental conditions, up to the development of hepatic coma, resulting in a significant burden on the health care system and a pronounced decline in quality of life. It has been reported that overt HE occurs in 30–40%, and MHE in 20–80%, of all LC cases [6].

Pharmacotherapy for HE includes primarily non-absorbable disaccharides such as lactulose and antibiotics such as rifaximin-α. Other agents are also used for HE therapy, for example, branched-chain oral amino acids (isoleucine, valine and leucine), L-ornithine-L-aspartate (LOLA), probiotics and some others [7].

Rifaximin-α has been studied for the treatment of HE in a number of comparative studies with placebo, other antibiotics and non-absorbable disaccharides. Those studies showed an effect of rifaximin-α that was equivalent or superior to the comparators and was well tolerated. In studies aimed at studying the concentration of ammonia in the blood with rifaximin-α, it was shown that rifaximin-α reduces the severity of HE and reduces the concentration of ammonia in serum by approximately 50% [3].

A number of studies have shown that long-term rifaximin-α therapy for 6 months reduces the risk of a second HE episode and the frequency of HE-related hospitalizations compared to placebo [8]. Rifaximin was also shown to reduce the risk of recurrence of overt HE in patients with cirrhosis and ≥2 episodes of overt HE in the previous 6 months, with HE episodes occurring in 22.1% of patients in the rifaximin group vs. 45.9% in the placebo group [9]. In addition, there is evidence that the use of rifaximin-α for 6 months compared to placebo reduced the relative risk of any first LC complication in patients with a MELD ≥ 12 and an international normalized ratio (INR) ≥ 1.2 [10].

The data on studying the effect of rifaximin-α on the microbiota and on the pathogenetic factors of PE are interesting. After an 8-week course of rifaximin-α, there was a significant increase in the level of long-chain fatty acids and an improvement in cognitive function with a slight change in the composition of the microbiota. The improvement in cognitive function is attributed to the effect of rifaximin-α on changes in microbiota-related metabolic function [11].

However, data on the efficiency of various regimens of rifaximin-α are still scarce. In particular, a small number of studies on quality of life during the treatment of patients with MHE are noteworthy. In this regard, the effect of rifaximin-α has recently been actively studied not only on overt HE, but also on MHE in patients with HC. Taking into account the available data, it seems advisable to study in more depth the effect of rifaximin-α on the quality of life of patients with LC and MHE to develop optimal methods and regimens for its use in such patients.

The main objective of this study was to evaluate the effect of rifaximin-α on the quality of life of patients with LC and MHE when used for 12 months. At the same time, this study examined the demographic characteristics of patients with HC, assessed symptoms associated with HE, and laboratory findings.

Thus, this study was designed to solve one of the urgent medical problems—optimization of therapeutic and prophylactic approaches in LC with MHE and was aimed at finding the optimal regimens for the use of the well-studied rifaximin-α and studying its effect on the main clinical and laboratory parameters and the quality of life of patients during long-term treatment.

## 2. Material and Methods

This was a multicenter, prospective, open-label, post-marketing observational study (NORMIND) conducted using electronic case report forms based on digital platform enrollme.ru.

This study involved 39 physicians from 19 cities in Russia. Medical investigators screened patients according to inclusion/exclusion criteria, obtained informed consent of the patient and enrolled the eligible patients in the study.

This study included 288 patients with LC and MHE of both sexes over the age of 18, who were prescribed a course of treatment with rifaximin-α by the attending physician in accordance with the product label and personal experience of the attending physician. The course of treatment did not depend on the study, and was determined independently by the physician.

Statistical analysis was performed on a population of patients who fully met the requirements of the protocol and completed all study visits (*n* = 258). A total of 30 patients did not complete the study: 15 patients due to death; 3 patients underwent liver transplantation; 1 patient withdrew from this study due to pregnancy; 11 patients withdrew without explanation.

Two subgroups were identified according to the extent of treatment completeness. Subgroup 1 *(n* = 41) (continuous course) included patients with 360 or more days of study treatment per year. In a continuous course, rifaximin-α was administered at a daily dose of 1200 mg.

Subgroup 2 (*n* = 217) (cyclic course) included patients that were administered the drug product less than 360 days per year. In a cyclic course, rifaximin-α was used for 7–14 days of each month in a daily dose of 600–1200 mg.

This study included screening and four visits (V), including V0 (screening) and V1–V4 (observation). Schematically, the study design is presented in Figure 1.

## 3. Inclusion/Exclusion Criteria

The inclusion criteria for the studies were as follows:Patients of either sex over the age of 18 years;Established LC and MHE diagnosis. MHE was diagnosed if the patient needed from 41 to 60 s to perform a number connection test (NCT) in the absence of changes in the neurological status;The patient was prescribed a course of rifaximin-α prior to enrollment in the study;Availability of the patient’s signed informed consent to the inclusion in the study and to personal data processing.

The patient could not be included in the study or should have been excluded from the study if he/she met at least one of the following exclusion criteria:Contraindications for the use of rifaximin-α;Causes of HE other than LC;History of an overt HE episode;Malignancies;Surgery scheduled for the study period (any);Pregnancy, breastfeeding or fertile women not using contraception;Current participation in another clinical study or in the last 30 days;Any other medical and non-medical reasons that, in the opinion of the physician, may prevent the patient from participating in the study.

In accordance with the Study Protocol, the following data were adopted as primary data for analysis:Demographic and anthropometric indicators:
Age;Sex;Height;Weight;
Primary endpoints:
Change in CLDQ (Chronic Liver Disease Questionnaire) total score over 12 months of follow-up;Secondary endpoints:
Study of rifaximin-α use in patients during 12 months of observation (number of courses of treatment, duration of treatment, average daily dose);Study of changes in the number connection test performance during 12 months of observation;Study of changes in the MHE symptom scale during 12 months of observation.

## 4. Statistical Analyses

Primary and secondary analysis is represented by descriptive statistics. All continuous variables were summed using the following parameters: n (sample size of available patients), mean, standard deviation, median, 25 and 75 percentiles, maximum and minimum.

The critical *p*-value and confidence intervals were calculated as two-sided. This study adopted a statistical significance level of 0.05 (two-sided testing, all *p*-values rounded to three decimal places).

Arithmetic mean, standard deviation, 95% confidence intervals, median, upper and lower quartiles were used to describe the continuous variables.

Categorical variables are presented as frequency percentages.

Substitution and imputation of missing data is not provided. All variables were compared before and after a specified observation period. To test the significance of differences, normally distributed data, the corresponding varieties of ANOVA repeated measures were used. In the case of other distributions, the Wilcoxon test was used.

Chi-square test or Fisher’s exact test were used to test the significance of categorical differences.

The study was approved by the decision of the Independent Interdisciplinary Committee for Ethical Expertise of Clinical Trials on 11 June 2021.

## 5. Results

Analysis of demographic data showed (Table 1) that both subgroups were predominantly female (58.5% in subgroup 1 and 54.4% in subgroup 2).

Table 2 presents patient data depending on anthropometric parameters. The data obtained indicate that, in subgroup 1, the average age of patients was 44.73 years, while in subgroup 2 it was 54.20 years. The mean height in both subgroups was not significantly different (172.88 subgroup 1 and 169.45 subgroup 2). The weight differed significantly, since the mean weight of patients in subgroup 1 was 69.44 kg, and in subgroup 2 80.88 kg.

Weight loss may be one of the concomitant conditions associated with liver cirrhosis. At baseline, at Visit 1, the weight of patients varied greatly from 42 to 132 kg. The data in the subgroups were also not similar. On average, the weight of the patients treated for 360 days during the year (subgroup 1 (continuous regimen)) was about 10 kg less in the cyclic subgroup. At the same time, it is noteworthy that the weight of patients on continuous regimen increased during the study, but not in subgroup 2, in which the mean weight was almost unchanged.

However, any changes in weight, both in the general population and in subgroups, did not reach statistical significance, on the basis of which it can be concluded that, during the year of observation, the weight of patients did not change significantly.

### 5.1. Quality of Life Assessments

The Chronic Liver Disease Questionnaire (CLDQ) is a tool for determining the quality of life in patients with chronic liver disease. In addition to measuring physical and mental health, the scale takes into account specific functional areas specific to patients with chronic liver disease. The final version of the scale contains 29 questions for six groups of indicators, including abdominal symptoms (questions 1, 5, 17), weakness (questions 2, 4, 8, 11, 13), systemic symptoms (questions 3, 6, 21, 23, 27), activity (questions 7, 9, 14), emotional function (questions 10, 12, 15, 16, 19, 20, 24, 26) and anxiety (questions 18, 22, 25, 28, 29). For each question, a Likert score ranging from 1 (greatest impairment) to 7 (least impairment) is used for quantitative assessment. The total score is calculated as the result of dividing the sum of the scores for each group of indicators by the number of questions in the group. The final score is obtained by dividing the sum of the scores by the total number of questions (*n* = 29). An increase in the mean value indicates an improvement in the indicator and the final score of the patient’s quality of life.

When conducting the study, the questionnaire was filled out by the physicians according to oral information provided by the patients at all visits (1–4). Only the final score was subject to statistical analysis. The results of the analysis of the CLDQ questionnaire are presented in Figure 2.

The analysis showed that the mean final score of the CLDQ questionnaire in the general population significantly increased. Pairwise analysis in the general population showed significance for all pairs of visits (*p* < 0.05). Significant changes were observed in both subgroups.

### 5.2. Psychometric Tests

The number connection test (NCT) is the most widely used test in psychometric studies to evaluate patients with HC. The test enables evaluation of the patient’s visual and spatial orientation and the psychomotor speed; when performing the test, the patient needs to sequentially connect the numbers from 1 to 25 printed scattered on a sheet of paper. The number connection test (NCT) was performed at each visit; time spent by the patient to perform the test was used to stage HE (Table 3).

At Visit 1, there was no statistically significant difference in the NCT performance time between the subgroups. Subsequently, both in the general population and in the subgroups, a decrease in the mean NCT performance was observed, and in the subgroup 1 (continuous regimen), the decrease was more pronounced than in the subgroup 2 (cyclic regimen). Changes in the score both in the general population and in the subgroups were statistically significant (*p* < 0.05).

To determine the differences in the subgroups, a comparison of the mean test performance time difference between Visit 4 and Visit 1 in each subgroup was made. In the continuous regimen subgroup, the time to complete the test decreased by 9.02 s (from 52.83 ± 5.00 s to 43.80 ± 5.84), while in the cyclic regimen subgroup, the time to complete the test decreased by 7.03 s (from 52.67 ± 5.20 to 45.64 ± 8.89). The differences between the subgroups were statistically significant. As follows from the data obtained, at Visit 1, the general population was homogeneous in terms of the HE stage calculated on the basis of NCT. At subsequent visits, the HE stage changed. Thus, there was a small (up to 2.7% in the general population) number of patients with a more severe HE stage (I, I–II), but changes mainly indicated a less severe HE stage or the absence of HE (up to 28.7% in the general population). It is also noteworthy that, in subgroup 1, this trend was more pronounced (observed in 43.9% of patients) than in subgroup 2 (25.8% of patients). The significance of the differences is confirmed (*p* < 0.05). Based on the above, it is possible to accept the significance of the observed changes in the NCT scores during the observations.

### 5.3. Dynamics of the Severity of Liver Cirrhosis According to Child–Turcotte–Pugh

Child–Turcotte–Pugh classification is used to assess the severity of LC and patient survival. The LC severity is assessed by the sum of points from 1 to 3 for each of the 5 or 6 parameters. A lower score indicates an improvement, and a lower disease class indicates a better survival prognosis. During the study, the estimates required to calculate scores and determine the LC class were recorded at each visit (Figure 3).

In the analysis of the LC classes frequency distribution depending on the observation visit determined by the Child–Turcotte–Pugh classification, an increase in the proportion of patients with Class A from 65.5% to 78.3% was observed in the general population (*p* < 0.05). At the same time, pronounced heterogeneity in the subgroups at Visit 1 was observed. In the comparative analysis, an increase in the proportion of patients with Class A was observed in the subgroups: in subgroup 1 from 24.4% to 31.7%, in subgroup 2 from 73.3% to 87.1%, at Visits 1 and 4, respectively. However, in subgroup 1, this change was not significant as opposed to subgroup 2.

### 5.4. Dynamics of Hyperammonemia

Blood ammonia levels were assessed at each visit during the study. Ammonia values by visit are shown in Figure 4.

The analysis showed that, in the general population, there was a significant decrease in the blood ammonia level. Pairwise analysis showed significance for pairs of visits 1–3, 1–4, and 2–4 (*p* < 0.05). When analyzed in subgroups, the significance of changes was demonstrated only in subgroup 1 for pairs 1–4 and 2–4 (*p* < 0.05). Analysis of the differences in ammonia levels at Visits 4 and 1 showed no significant differences between the subgroups.

### 5.5. Dynamics of Hepatic Encephalopathy Symptoms

The study evaluated the manifestations of HE symptoms on a 4-point Likert scale from 0 to 3, where 0 is no symptoms, 1 is mild, 2 is moderate, 3 is severe. The following symptoms were assessed: impaired vision, impaired concentration and memory, cognitive impairment, clouded sensorium, decreased performance, decreased sensitivity, and irritability. A higher score correlated with a more severe manifestation of the symptom, so a decrease in the mean score showed an improvement in the patient’s condition, according to the physician’s assessment. The dynamics of concentration and memory impairment depending on the rifaximin-a regimen and the observation period are shown in Figure 5.

During the study, it was shown that the mean concentration and memory impairment score in the general population decreased significantly. Pairwise analysis showed significant difference for all pairs of visits (*p* < 0.05). When analyzed in subgroups, the significance of changes was revealed in both subgroups. Analysis of the differences in scores at Visits 4 and 1 showed no significant differences between the subgroups (*p* < 0.05).

Figure 6 shows the cognitive impairment score dynamics.

During the study, it was shown that the mean cognitive impairment score in the general population decreased significantly. Pairwise analysis showed significance for all pairs of visits (*p* < 0.05). Significant changes were observed in both subgroups.

Analysis of the differences in scores at Visits 4 and 1 showed no significant differences between the subgroups (*p* < 0.05).

Decreased performance score is shown in Figure 7.

The data obtained indicate that the mean performance score in the general population decreased significantly. Pairwise analysis showed significance for all pairs of visits (*p* < 0.05). Significance of changes was observed in both subgroups for all pairs of visits (*p* < 0.05). The mean performance score from Visit 1 to Visit 4 decreased in subgroup 1 from 1.29 ± 0.72 to 0.17 ± 0.44 and in subgroup 2 from 1.27 ± 0.75 to 0.56 ± 0.60. Analysis of the differences in scores at Visits 4 and 1 showed no significant differences between the subgroups (*p* < 0.05).

Analysis of such symptoms associated with HE as visual impairment, clouded sensorium, decreased sensitivity, and irritability showed a significant decrease in their severity at Visit 4 compared to Visit 1 both in the general population and by subgroups without statistical significance of differences between subgroups (Table 4).

## 6. Discussion

It is known that rifaximin-α is hardly absorbed in the gastrointestinal tract. However, LC can significantly affect the pharmacokinetics of any medicine, including the study drug, and systemic absorption in these patients is markedly increased compared to the control group. Thus, plasma concentrations of rifaximin-α up to 10 ng/mL were observed in patients with LC compared to only 1 ng/mL in the control group. This requires careful examination, especially when daily long-term therapy for chronic liver diseases, such as LC with MHE, is proposed [12].

Various studies have shown that the use of rifaximin-α reduced ammonia levels, improved the results of psychometric tests and improved cognitive function and mental status in patients with LC and an acute HE episode [13]. A systematic review demonstrated that rifaximin-α was at least as efficient or superior to non-absorbable disaccharides and antibacterial drugs in alleviating signs or symptoms observed in patients with mild to moderate HE [14].

A number of studies have investigated the effect of rifaximin-α on quality of life in patients with LC and HE. For example, one study showed that patients with MHE who received rifaximin-α for 8 weeks showed significantly greater improvement in psychosocial aspects, as well as in driving skills and cognitive function, than patients who received placebo [15]. These results were confirmed in another randomized controlled trial in which the authors demonstrated that rifaximin-α significantly improved both cognitive function and human quality of life in patients with MHE [16].

The NORMIND study aimed to investigate the effect of long-term (12 months) rifaximin-α treatment in patients with LC and MHE on quality of life, a number of some objective and subjective disease symptoms, as well as laboratory parameters. Data were analyzed for the general population who fully completed the study at all four visits (*n* = 258). Additionally, two subgroups were retrospectively identified: subgroup 1—patients received rifaximin-α for at least 360 days per year (*n* = 41)—and subgroup 2—patients received rifaximin-α cyclically from course to course (less than 360 days) (*n* = 217). The purpose of dividing into the subgroups was to assess possible differences in the effects of continuous and cyclic rifaximin-α regimens. At the same time, it should be noted that, perhaps, the differences in the assignment of treatment courses were dictated by the characteristics of the course of the disease and, therefore, may not objectively reflect the effect of the therapy.

The profile of patients with MHE, including anthropometric parameters, was studied during the study. It is well known that weight can decrease in patients with LC. However, in our study, no negative weight trend was observed and the mean weight of the patients had no significant decrease over the entire follow-up period.

As per the CLDQ questionnaire, there was a significant increase in the mean total score from Visit 1 to Visit 4 in both the general population and in subgroup 1 and 2 without differences between subgroups.

The number connection test (NCT), which assesses the visual–spatial orientation and psychomotor speed in patients with HE, showed a significant decrease in the performance time during the observation period for both the general population and for each of the subgroups. With continuous therapy, the dynamics of the decrease in test performance time was more pronounced compared to cyclic therapy.

According to the results of the analysis of the distribution of patients, according to the Child–Turcotte–Pugh classification, in the general population, the mean score significantly decreased from the first to the last visit. At the same time, there was a pronounced heterogeneity at the inclusion visit, since, in the continuous therapy subgroup, there were more severe patients compared to the cyclic therapy subgroup.

The Likert scale, which includes a score of 4 points (from 0—minimum manifestation—to 3—maximum manifestation), was used to assess the physician’s assessment of a number of symptoms and manifestations of HE, such as: impaired vision, concentration and memory impairment, cognitive impairment, and clouded sensorium. There was a clear and significant decrease in the mean score in both the general population and in subgroup 1 and 2. For the subgroup of continuous regimen, a more significant decrease in the severity of symptoms such as concentration and memory impairment, cognitive impairment, decreased performance is shown.

## 7. Conclusions

All study parameters, including quality of life, number connection test, symptoms associated with MHE, blood ammonia level showed a positive trend in the general population that completed the study according to the protocol.

According to the results of statistical analysis, the distribution of the population into subgroups according to the duration of the course of treatment showed a difference between the subgroups in a more pronounced decrease in the time to perform the number connection test, a decrease in the severity of symptoms such as concentration and memory impairment, cognitive impairment, clouded sensorium, and decreased performance, which may indicate the benefit of continuous administration of rifaximin-α for patients with liver cirrhosis and MHE.

## Figures and Tables

**Figure 1 diagnostics-13-03239-f001:**
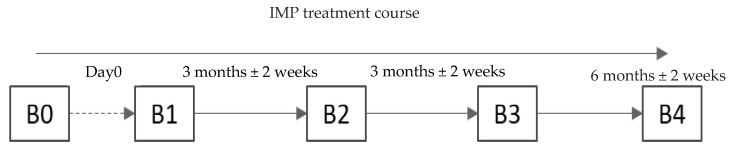
Study design. The screening visit could have been performed on the day the patient was administered the study treatment (Visit 1). Visit 2 was conacducted 3 months ± 2 weeks after Visit 1. Visit 3 was conducted 3 months ± 2 weeks after Visit 2. Visit 4 was conducted 6 months ± 2 weeks after Visit 3.

**Figure 2 diagnostics-13-03239-f002:**
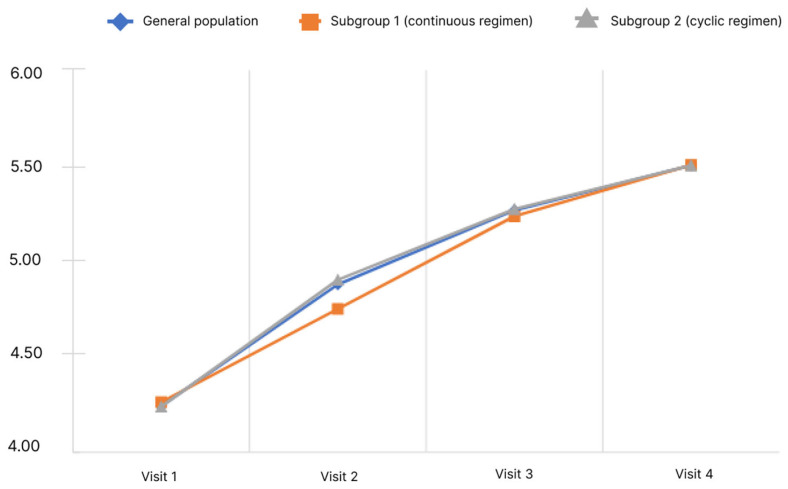
Dynamics of CLDQ indicators depending on the subgroup and observation period.

**Figure 3 diagnostics-13-03239-f003:**
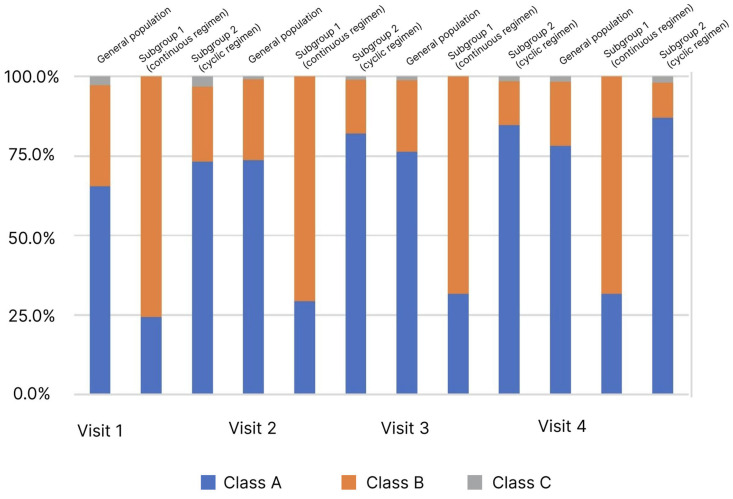
Distribution of patients (%) by stage of liver cirrhosis (Child–Turcotte–Pugh classification) and observation period.

**Figure 4 diagnostics-13-03239-f004:**
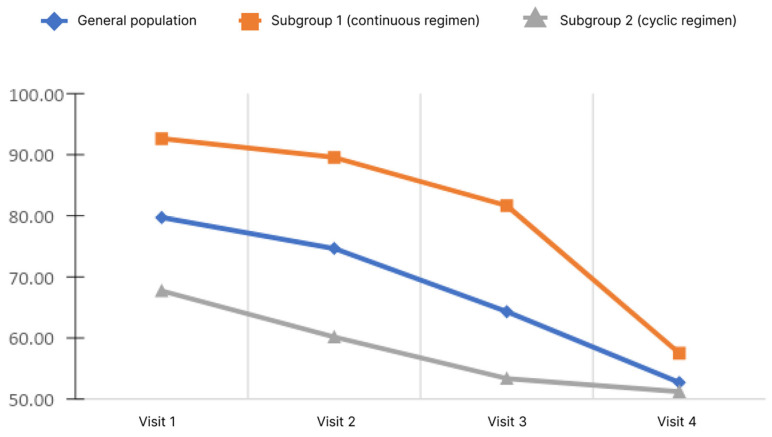
Changes of blood ammonia level depending on the subgroup and observation period.

**Figure 5 diagnostics-13-03239-f005:**
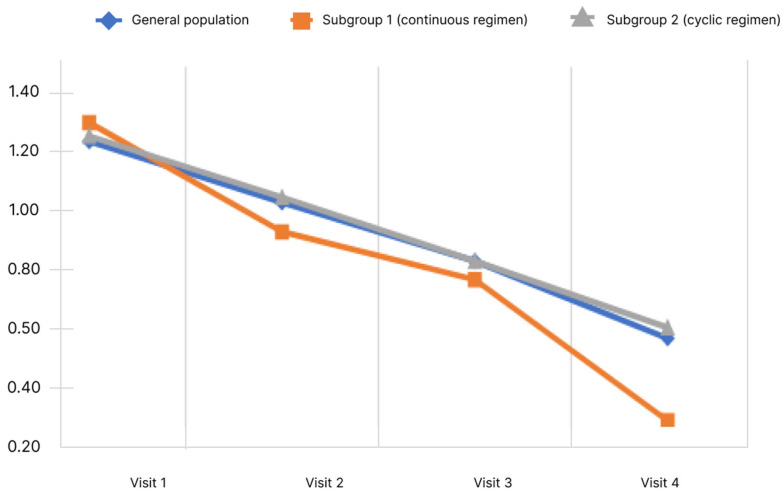
Changes of concentration level and memory impairment score depending on the subgroup and observation period.

**Figure 6 diagnostics-13-03239-f006:**
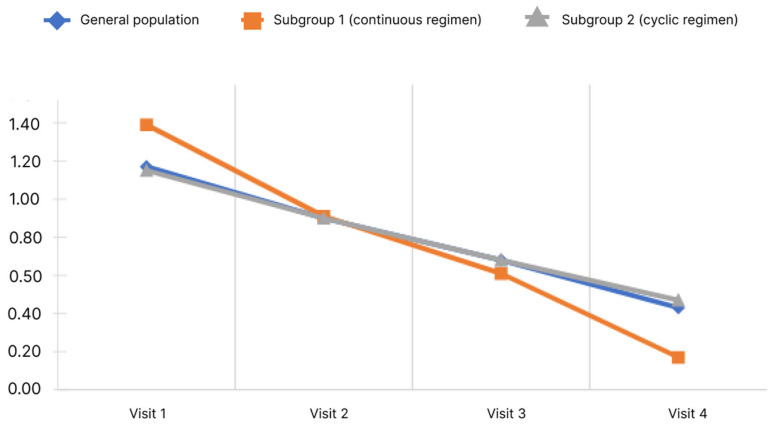
Changes of cognitive impairment score depending on the subgroup and observation period.

**Figure 7 diagnostics-13-03239-f007:**
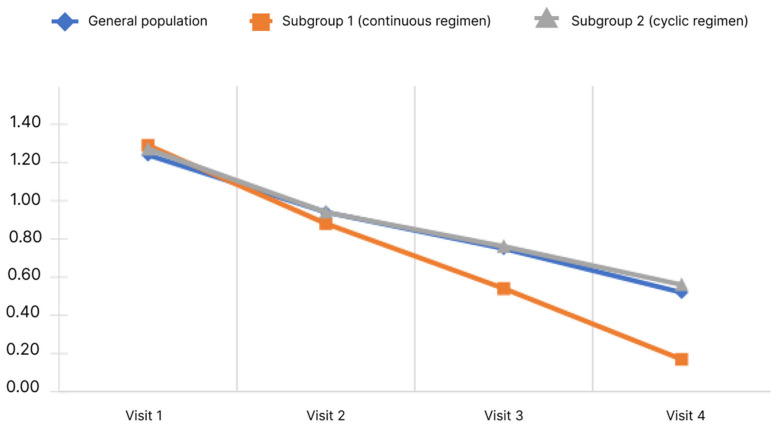
Changes of decreased performance scores depending on the subgroup and observation period.

**Table 1 diagnostics-13-03239-t001:** Patient distribution based on demographic data.

All Patients who Completed per-Protocol Visits		
n	258	
Male	116	45.0%
Female	142	55.0%
Subgroup 1 (continuous regimen)		
n	41	
Male	17	41.5%
Female	24	58.5%
Subgroup 2 (cyclic regimen)		
n	217	
Male	99	45.6%
Female	118	54.4%

**Table 2 diagnostics-13-03239-t002:** Patient distribution based on anthropometric data.

All Patients Who Completed per-Protocol Visits			
Sample size	258	258	258
Mean value	52.70	169.99	79.07
Standard deviation	12.57	8.19	17.04
Median	52.00	170.00	79.00
25th quartile	44.00	164.00	67.00
75th quartile	61.00	176.00	92.00
Minimum	19.00	150.00	42.00
Maximum	83.00	190.00	132.00
Subgroup 1 (continuous regimen)			
Sample size	41	41	41
Mean value	44.73	172.88	69.44
Standard deviation	12.43	9.18	13.93
Median	43.00	173.00	70.00
25th quartile	37.00	165.00	59.00
75th quartile	53.00	179.00	80.00
Minimum	23.00	151.00	43.00
Maximum	75.00	189.00	108.00
Subgroup 2 (cyclic regimen)			
Sample size	217	217	217
Mean value	54.20	169.45	80.88
Standard deviation	12.04	7.89	16.98
Median	54.00	170.00	82.00
25th quartile	46.00	164.00	68.00
75th quartile	62.00	175.00	94.00
Minimum	19.00	150.00	42.00
Maximum	83.00	190.00	132.00

**Table 3 diagnostics-13-03239-t003:** Distribution of patients depending on the stage of hepatic encephalopathy and the observation period (according to the results of the number connection test).

HE Stage	Visit 1		Visit 2		Visit 3		Visit 4	
	n	%	n	%	n	%	n	%
General population								
n	258		258		258		258	
No	0	0.0%	21	8.1%	36	14.0%	74	28.7%
0–I	258	100.0%	230	89.1%	216	83.7%	178	69.0%
I, I–II	0	0.0%	7	2.7%	5	1.9%	6	2.3%
II	0	0.0%	0	0.0%	1	0.4%	0	0.0%
II–III	0	0.0%	0	0.0%	0	0.0%	0	0.0%
Subgroup 1 (continuous regimen)								
n	41		41		41		41	
No	0	0.0%	2	4.9%	7	17.1%	18	43.9%
0–I	41	100.0%	38	92.7%	32	78.0%	23	56.1%
I, I–II	0	0.0%	1	2.4%	2	4.9%	0	0.0%
II	0	0.0%	0	0.0%	0	0.0%	0	0.0%
II–III	0	0.0%	0	0.0%	0	0.0%	0	0.0%
Subgroup 2 (cyclic regimen)								
n	217		217		217		217	
No	0	0.0%	19	8.8%	29	13.4%	56	25.8%
0–I	217	100.0%	192	88.5%	184	84.8%	155	71.4%
I, I–II	0	0.0%	6	2.8%	3	1.4%	6	2.8%
II	0	0.0%	0	0.0%	1	0.5%	0	0.0%
II–III	0	0.0%	0	0.0%	0	0.0%	0	0.0%

Scale for converting NCT results to the HE stage: <40 s: no HE; 41–60 s: HE stage 0-I; 61–90 s: HE stage I, I–II; 91–120 s: HE stage II; >120 s: HE stage II–III.

**Table 4 diagnostics-13-03239-t004:** Scores and dynamics between Visit 4 and Visit 1 for visual impairment, clouded sensorium, decreased sensitivity, and irritability in the general population of patients who completed the study according to the protocol.

	V1	V4	V4–V1	*p*
Visual impairment				
Mean	0.84	0.56	−0.28	<0.05
SD	0.64	0.49	0.60	
Clouded sensorium				
Mean	1.27	0.51	−0.76	<0.05
SD	0.74	0.59	0.91	
Decreased sensitivity				
Mean	0.78	0.24	−0.54	<0.05
SD	0.77	0.45	0.82	
Irritability				
Mean	1.26	0.47	−0.79	<0.05
SD	0.86	0.55	0.99	

## Data Availability

Not applicable.

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
