# Peer review of "Comparative Analysis of the Efficacy of Different Regimens of 12 Months Rifaximin-Alfa Therapy in Patients with Liver Cirrhosis and Minimal Hepatic Encephalopathy"

_diagnostics, 2023, doi:10.3390/diagnostics13203239_

Round 1

Reviewer 1 Report

The manuscript analyzed the eficacy rifaximin-alpha therapy  in patients with liver cirrhosis and minimal hepatic encephalopathy.The manuscript is well conceived, the abstract and conclusions are well written, the images and diagrams look good.

But I have a few questions for the authors.The question of the uniformity of the results (258 patients from 19 cities in Russia were included, and the research was conducted by 39 researchers)? Did power analysis perform when assessing the optimal sample size?

Author Response

The study was non-interventional and did not initially include a control group, and was designed to conduct a case series to study the effect of treatment with rifaximin-α on health outcomes in patients with minimal hepatic encephalopathy. The research hypothesis assumed that a positive effect would be observed, but at the research preparation stage, quantification of this hypothesis was not available. Therefore, no formal sample size calculation was performed. It was empirically assumed that observation of 200 patients included in the study would provide the amount of necessary data sufficient for reliable statistical processing of the results.

Reviewer 2 Report

An important observational study about the long-term administration of rifaximin. Congratulations for the clinicians organized in that large multicenter study! I would like to suggest to follow the cohort of the cirrhotic patients included in that broad database in the future. The addition of information, in my view, is recommendable:

1. Complex properties of rifaximin, besides antibacterial activity

2. Novel mechanisms of hepatic encephalopathy in the "Introduction" part

3. Clarification of "general population" used to compare subgroup 1 and 2 in the "Methods"

4. EASL (or other) consensus guidelines on the management of liver encephalopathy in the referred literature.

Author Response

Thank you for your interest in our work!
All your comments were taken into account and added to the manuscript, highlighted in yellow (see attachment).
Answering question 3, I would like to note that the article indicated this information, namely:
"Statistical analysis was performed on a population of patients who fully met the requirements of the protocol and completed all study visits (n=258). 30 patients did not complete the study: 15 patients due to death; 3 patients underwent liver transplantation; 1 patient withdrew from the study due to pregnancy; 11 patients withdrew without explanation.
All patients included in the statistical analysis (n=258) were divided into subgroup 1 (n=41) and subgroup 2 (n=217) depending on the course of rifaximin during follow-up.
Subgroup 1 (n=41) (continuous course) included patients with 360 or more days of study treatment per year. In a continuous course, rifaximin-α was administered at a daily dose of 1200 mg.
Subgroup 2 (n=217) (cyclic course) included patients that were administered the drug product less than 360 days per year. In a cyclic course, rifaximin-α was used for 7–14 days of each month in a daily dose of 600–1200 mg"

Reviewer 3 Report

Bakulin  et  al. aimed  to to evaluate the effect of rifaximin-α on the quality of life of patients with LC and MHE when used for 12 months. The scientific study was well designed. The study involved 39 physicians from 19 cities of Russia. The study included 288 patients with LC and MHE of both sexes over the age of 18,who were prescribed a course of treatment with rifaximin-α by the attending physician in accordance with the product label and personal experience of the attending physician. Two subgroups were identified according to the extent of treatment completeness. Subgroup 1 (n=41) (continuous course) included patients with 360 or more days of study  treatment per year. Subgroup 2 (n=217) (cyclic course) included patients that were administered the drug  product less than 360 days per year. In a cyclic course, rifaximin-α was used for 7–14 days  of each month in a daily dose of 600–1200 mg. According to the results of the analysis of the distribution of patients, according to  the Child-Turcotte-Pugh classification, in the general population, the mean score signifi-  cantly decreased from the first to the last visit. For the subgroup of continuous regimen, a more significant decrease  in the severity of symptoms such as concentration and memory impairment, cognitive  impairment, decreased performance was shown. The study  confirmed the data previously obtained through retrospective studies with a prospective study.  The  authors  also contributed by providing useful data regarding the type of treatment with rifaximin. Thank  you  for  giving  opportunity  to  review  this  study.

Author Response

Thank you very much for the time you devoted to our work!